# Forward-Backward Sweep Method for the System of HJB-FP Equations in Memory-Limited Partially Observable Stochastic Control

**DOI:** 10.3390/e25020208

**Published:** 2023-01-21

**Authors:** Takehiro Tottori, Tetsuya J. Kobayashi

**Affiliations:** 1Department of Mathematical Informatics, Graduate School of Information Science and Technology, The University of Tokyo, Tokyo 113-8654, Japan; 2Institute of Industrial Science, The University of Tokyo, Tokyo 153-8505, Japan; 3Department of Electrical Engineering and Information Systems, Graduate School of Engineering, The University of Tokyo, Tokyo 113-8654, Japan; 4Universal Biology Institute, The University of Tokyo, Tokyo 113-8654, Japan

**Keywords:** decision-making, optimal control, stochastic control, incomplete information, memory limitation, mean-field control

## Abstract

Memory-limited partially observable stochastic control (ML-POSC) is the stochastic optimal control problem under incomplete information and memory limitation. To obtain the optimal control function of ML-POSC, a system of the forward Fokker–Planck (FP) equation and the backward Hamilton–Jacobi–Bellman (HJB) equation needs to be solved. In this work, we first show that the system of HJB-FP equations can be interpreted via Pontryagin’s minimum principle on the probability density function space. Based on this interpretation, we then propose the forward-backward sweep method (FBSM) for ML-POSC. FBSM is one of the most basic algorithms for Pontryagin’s minimum principle, which alternately computes the forward FP equation and the backward HJB equation in ML-POSC. Although the convergence of FBSM is generally not guaranteed in deterministic control and mean-field stochastic control, it is guaranteed in ML-POSC because the coupling of the HJB-FP equations is limited to the optimal control function in ML-POSC.

## 1. Introduction

In many practical applications of the stochastic optimal control theory, several constraints need to be considered. In the cases of small devices [1,2] and biological systems [3,4,5,6,7,8], for example, incomplete information and memory limitation become predominant because their sensors are extremely noisy and their memory resources are severely limited. To take into account one of these constraints, incomplete information, partially observable stochastic control (POSC) has been extensively studied in the stochastic optimal control theory [9,10,11,12,13]. However, because POSC cannot take into account the other constraint, memory limitation, it is not practical enough for designing memory-limited controllers for small devices and biological systems. To resolve this problem, memory-limited POSC (ML-POSC) has recently been proposed [14]. Because ML-POSC formulates noisy observation and limited memory explicitly, ML-POSC can take into account both incomplete information and memory limitation in the stochastic optimal control problem.

However, ML-POSC cannot be solved in a similar way as completely observable stochastic control (COSC), which is the most basic stochastic optimal control problem [15,16,17,18]. In COSC, the optimal control function depends only on the Hamilton–Jacobi–Bellman (HJB) equation, which is a time-backward partial differential equation given a terminal condition (Figure 1a) [15,16,17,18]. Therefore, the optimal control function of COSC can be obtained by solving the HJB equation backward in time from the terminal condition, which is called the value iteration method [19,20,21]. In contrast, the optimal control function of ML-POSC depends not only on the HJB equation but also on the Fokker–Planck (FP) equation, which is a time-forward partial differential equation given an initial condition (Figure 1b) [14]. Because the HJB equation and the FP equation interact with each other through the optimal control function in ML-POSC, the optimal control function of ML-POSC cannot be obtained by the value iteration method.

To propose an algorithm to solve ML-POSC, we first show that the system of HJB-FP equations can be interpreted via Pontryagin’s minimum principle on the probability density function space. Pontryagin’s minimum principle is one of the most representative approaches to the deterministic optimal control problem, which converts it into the two-point boundary value problem of the forward state equation and the backward adjoint equation [22,23,24,25]. We formally show that the system of HJB-FP equations is an extension of the system of adjoint and state equations from the deterministic optimal control problem to the stochastic optimal control problem.

The system of HJB-FP equations also appears in the mean-field stochastic control (MFSC) [26,27,28]. Although the relationship between the system of HJB-FP equations and Pontryagin’s minimum principle has been briefly mentioned in MFSC [29,30,31], its details have not yet been investigated. In this work, we investigate it in more detail by deriving the system of HJB-FP equations in a similar way to Pontryagin’s minimum principle. We note that our derivations are formal, not analytical, and more mathematically rigorous proofs remain future challenges. However, our results are consistent with many conventional results and also provide a useful perspective in proposing an algorithm.

We then propose the forward-backward sweep method (FBSM) for ML-POSC. FBSM is an algorithm to compute the forward FP equation and the backward HJB equation alternately, which can be interpreted as an extension of the value iteration method. FBSM has been proposed in Pontryagin’s minimum principle of the deterministic optimal control problem, which computes the forward state equation and the backward adjoint equation alternately [32,33,34]. Because FBSM is easy to implement, it has been used in many applications [35,36]. However, the convergence of FBSM is not guaranteed in deterministic control except for special cases [37,38] because the coupling of adjoint and state equations is not limited to the optimal control function (Figure 1c). In contrast, we show that the convergence of FBSM is generally guaranteed in ML-POSC because the coupling of the HJB-FP equations is limited only to the optimal control function (Figure 1b).

FBSM is called the fixed-point iteration method in MFSC [39,40,41,42]. Although the fixed-point iteration method is the most basic algorithm to solve MFSC, its convergence is not guaranteed for the same reason as deterministic control (Figure 1d). Therefore, ML-POSC is a special and nice class of optimal control problems where FBSM or the fixed-point iteration method is guaranteed to converge.

This paper is organized as follows: In Section 2, we formulate ML-POSC. In Section 3, we derive the system of HJB-FP equations of ML-POSC from the viewpoint of Pontryagin’s minimum principle. In Section 4, we propose FBSM for ML-POSC and prove its convergence. In Section 5, we apply FBSM to the linear-quadratic-Gaussian (LQG) problem. In Section 6, we verify the convergence of FBSM by numerical experiments. In Section 7, we discuss our work. In Appendix A, we briefly review Pontryagin’s minimum principle of deterministic control. In Appendix B, we derive the system of HJB-FP equations of MFSC from the viewpoint of Pontryagin’s minimum principle. In Appendix C, we show the detailed derivations of our results.

## 2. Memory-Limited Partially Observable Stochastic Control

In this section, we briefly review the formulation of ML-POSC [14], which is the stochastic optimal control problem under incomplete information and memory limitation.

### 2.1. Problem Formulation

This subsection outlines the formulation of ML-POSC [14]. The state of the system $x_{t}\in R^{d_{x}}$ at time t∈[0,T] evolves by the following stochastic differential equation (SDE):(1)$$\begin{array}{cc} dx_{t} & =b\left(t,x_{t},u_{t}\right)dt+\sigma \left(t,x_{t},u_{t}\right)d\omega_{t}, \end{array}$$
where $x_{0}$ obeys $p_{0}\left(x_{0}\right)$, $u_{t}\in R^{d_{u}}$ is the control, and $\omega_{t}\in R^{d_{\omega }}$ is the standard Wiener process. In COSC [15,16,17,18], because the controller can completely observe the state $x_{t}$, it determines the control $u_{t}$ based on the state $x_{t}$ as $u_{t}=u\left(t,x_{t}\right)$. By contrast, in POSC [9,10,11,12,13] and ML-POSC [14], the controller cannot directly observe the state $x_{t}$ and instead obtains the observation $y_{t}\in R^{d_{y}}$, which evolves by the following SDE:(2)$$\begin{array}{cc} dy_{t} & =h\left(t,x_{t}\right)dt+\gamma \left(t\right)d\nu_{t}, \end{array}$$
where $y_{0}$ obeys $p_{0}\left(y_{0}\right)$, and $\nu_{t}\in R^{d_{\nu }}$ is the standard Wiener process. In POSC [9,10,11,12,13], because the controller can completely memorize the observation history $y_{0:t}:=\left\{y_{\tau }|\tau \in \left[0,t\right]\right\}$, it determines the control $u_{t}$ based on the observation history $y_{0:t}$ as $u_{t}=u\left(t,y_{0:t}\right)$. In ML-POSC [14], by contrast, because the controller cannot completely memorize the observation history $y_{0:t}$, it compresses the observation history $y_{0:t}$ into the finite-dimensional memory $z_{t}\in R^{d_{z}}$, which evolves by the following SDE:(3)$$\begin{array}{c} dz_{t}=c\left(t,z_{t},v_{t}\right)dt+\kappa \left(t,z_{t},v_{t}\right)dy_{t}+\eta \left(t,z_{t},v_{t}\right)d\xi_{t}, \end{array}$$
where $z_{0}$ obeys $p_{0}\left(z_{0}\right)$, $v_{t}\in R^{d_{v}}$ is the control, and $\xi_{t}\in R^{d_{\xi }}$ is the standard Wiener process. The memory dimension $d_{z}$ is determined by the available memory size of the controller. In addition, the memory noise $\xi_{t}$ represents the intrinsic stochasticity of the memory to be used. Therefore, unlike the conventional POSC, ML-POSC can explicitly take into account the memory size and noise of the controller. Furthermore, because the memory dynamics (Equation 3) depends on the memory control $v_{t}$, it can be optimized through the memory control $v_{t}$, which is expected to realize the optimal compression of the observation history $y_{0:t}$ into the limited memory $z_{t}$. In ML-POSC [14], the controller determines the state control $u_{t}$ and the memory control $v_{t}$ based on the memory $z_{t}$ as follows:(4)$$\begin{array}{c} u_{t}=u\left(t,z_{t}\right),\;v_{t}=v\left(t,z_{t}\right). \end{array}$$

The objective function of ML-POSC is given by the following expected cumulative cost function:(5)$$\begin{array}{c} J\left[u,v\right]:=E_{p(x_{0:T},y_{0:T},z_{0:T};u,v)}\left[\int_{0}^{T}f\left(t,x_{t},u_{t},v_{t}\right)dt+g\left(x_{T}\right)\right], \end{array}$$
where *f* is the cost function, *g* is the terminal cost function, $p(x_{0:T},y_{0:T},z_{0:T};u,v)$ is the probability of $x_{0:T}$, $y_{0:T}$, and $z_{0:T}$ given *u* and *v* as parameters, and $E_{p}\left[\cdot \right]$ is the expectation with respect to the probability *p*. Because the cost function *f* depends on the memory control $v_{t}$, ML-POSC can explicitly take into account the memory control cost, which is also impossible with the conventional POSC.

ML-POSC is the problem of finding the optimal state control function $u^{*}$ and the optimal memory control function $v^{*}$ that minimize the expected cumulative cost function J[u,v] as follows:(6)$$\begin{array}{c} u^{*},v^{*}:=\arg\min_{u,v}J\left[u,v\right]. \end{array}$$

ML-POSC first formulates the finite-dimensional and stochastic memory dynamics explicitly, then optimizes the memory control by considering the memory control cost. As a result, unlike the conventional POSC, ML-POSC is a practical framework for memory-limited controllers where the memory size, noise, and cost are imposed and non-negligible.

The previous work [14] has shown the validity and effectiveness of ML-POSC. In the LQG problem of conventional POSC, the observation history $y_{0:T}$ can be compressed into the Kalman filter without a loss of performance [10,18,43]. Because the Kalman filter is finite-dimensional, it can be interpreted as the finite-dimensional memory $z_{t}$ and discussed in terms of ML-POSC. The previous work [14] has proven that the optimal memory dynamics of ML-POSC become the Kalman filter in this problem, which indicates that ML-POSC is a consistent framework with the conventional POSC. Furthermore, the previous work [14] has demonstrated the effectiveness of ML-POSC in the LQG problem with memory limitation and in the non-LQG problem by numerical experiments.

### 2.2. Problem Reformulation

Although the formulation of ML-POSC in the previous subsection is intuitive, it is inconvenient for further mathematical investigations. To address this problem, we reformulate ML-POSC in this subsection. The formulation in this subsection is simpler and more general than that in the previous subsection.

First, we define an extended state $s_{t}$ as follows:(7)$$\begin{array}{c} s_{t}:=\left(\begin{array}{c} x_{t}\\ z_{t} \end{array}\right)\in R^{d_{s}}, \end{array}$$
where $d_{s}=d_{x}+d_{z}$. The extended state $s_{t}$ evolves by the following SDE:(8)$$\begin{array}{c} ds_{t}=\tilde{b}\left(t,s_{t},{\tilde{u}}_{t}\right)dt+\tilde{\sigma }\left(t,s_{t},{\tilde{u}}_{t}\right)d{\tilde{\omega }}_{t}, \end{array}$$
where $s_{0}$ obeys $p_{0}\left(s_{0}\right)$, ${\tilde{u}}_{t}\in R^{d_{\tilde{u}}}$ is the control, and ${\tilde{\omega }}_{t}\in R^{d_{\tilde{\omega }}}$ is the standard Wiener process. ML-POSC determines the control ${\tilde{u}}_{t}\in R^{d_{\tilde{u}}}$ based on the memory $z_{t}$ as follows:(9)$$\begin{array}{c} {\tilde{u}}_{t}=\tilde{u}\left(t,z_{t}\right). \end{array}$$
The extended state SDE (Equation 8) includes the previous SDEs (Equation 1)–(Equation 3) as a special case because they can be represented as follows:(10)$$\begin{array}{cc} \; & ds_{t}=\left(\begin{array}{c} b(t,x_{t},u_{t})\\ c\left(t,z_{t},v_{t}\right)+\kappa \left(t,z_{t},v_{t}\right)h\left(t,x_{t}\right) \end{array}\right)dt\\ \; & +\left(\begin{array}{ccc} \sigma (t,x_{t},u_{t}) & O & O\\ O & \kappa \left(t,z_{t},v_{t}\right)\gamma \left(t\right) & \eta (t,z_{t},v_{t}) \end{array}\right)\left(\begin{array}{c} d\omega_{t}\\ d\nu_{t}\\ d\xi_{t} \end{array}\right), \end{array}$$
where $p_{0}\left(s_{0}\right)=p_{0}\left(x_{0}\right)p_{0}\left(z_{0}\right)$.

The objective function of ML-POSC is given by the following expected cumulative cost function:(11)$$\begin{array}{c} J\left[\tilde{u}\right]:=E_{p(s_{0:T};\tilde{u})}\left[\int_{0}^{T}\tilde{f}\left(t,s_{t},{\tilde{u}}_{t}\right)dt+\tilde{g}\left(s_{T}\right)\right]. \end{array}$$
where $\tilde{f}$ is the cost function and $\tilde{g}$ is the terminal cost function. It is obvious that this objective function (Equation 11) is more general than that in the previous subsection (Equation 5).

ML-POSC is the problem of finding the optimal control function ${\tilde{u}}^{*}$ that minimizes the expected cumulative cost function $J[\tilde{u}]$ as follows:(12)$$\begin{array}{c} {\tilde{u}}^{*}:=\arg\min_{\tilde{u}}J\left[\tilde{u}\right]. \end{array}$$

In the following sections, we mainly consider the formulation of this subsection because it is simpler and more general than that in the previous subsection. Moreover, we omit $\tilde{\cdot }$ for simplicity of notation.

## 3. Pontryagin’s Minimum Principle

If the control $u_{t}$ is determined based on the extended state $s_{t}$ as $u_{t}=u\left(t,s_{t}\right)$, ML-POSC is the same problem with COSC of the extended state, and its optimality conditions can be obtained in the conventional way [15,16,17,18]. In reality, however, because ML-POSC determines the control $u_{t}$ based only on the memory $z_{t}$ as $u_{t}=u\left(t,z_{t}\right)$, its optimality conditions cannot be obtained in a similar way as COSC. In the previous work [14], the optimality conditions of ML-POSC were obtained by employing a mathematical technique of MFSC [30,31].

In this section, we obtain the optimality conditions of ML-POSC by employing Pontryagin’s minimum principle [22,23,24,25] on the probability density function space (Figure 2 (bottom right)). The conventional approach in ML-POSC [14] and MFSC [30,31] can be interpreted as a conversion from Bellman’s dynamic programming principle (Figure 2 (top right)) to Pontryagin’s minimum principle (Figure 2 (bottom right)) on the probability density function space.

In Appendix A, we briefly review Pontryagin’s minimum principle in deterministic control (Figure 2 (left)). In this section, we obtain the optimality conditions of ML-POSC in a similar way as Appendix A (Figure 2 (right)). Furthermore, in Appendix B, we obtain the optimality conditions of MFSC in a similar way as Appendix A (Figure 2 (right)). MFSC is more general than ML-POSC except for the partial observability. In particular, the expected Hamiltonian is non-linear with respect to the probability density function in MFSC, while it is linear in ML-POSC.

Although our derivations are formal, not analytical, and more mathematically rigorous proofs remain future challenges, our results are consistent with the conventional results of COSC [15,16,17,18], ML-POSC [14], and MFSC [26,27,28,30,31], and also provide a useful perspective in proposing an algorithm.

### 3.1. Preliminary

In this subsection, we show a useful result in obtaining Pontryagin’s minimum principle. Given arbitrary control functions *u* and $u^{\prime }$, $J\left[u\right]-J\left[u^{\prime }\right]$ can be calculated as follows:(13)$$\begin{array}{c} J\left[u\right]-J\left[u^{\prime }\right]=\int_{0}^{T}\left(E_{p(t,s)}\left[H(t,s,u,w^{\prime })\right]-E_{p(t,s)}\left[H(t,s,u^{\prime },w^{\prime })\right]\right)dt, \end{array}$$
where H is the Hamiltonian, which is defined as follows:(14)$$\begin{array}{c} H\left(t,s,u,w\right):=f\left(t,s,u\right)+L_{u}w\left(t,s\right). \end{array}$$
$L_{u}$ is the backward diffusion operator, which is defined as follows:(15)$$\begin{array}{cc} L_{u}w\left(t,s\right) & :=\sum_{i=1}^{d_{s}}b_{i}\left(t,s,u\right)\frac{\partial w(t,s)}{\partial s_{i}}+\frac{1}{2}\sum_{i,j=1}^{d_{s}}D_{ij}\left(t,s,u\right)\frac{\partial^{2}w\left(t,s\right)}{\partial s_{i}\partial s_{j}}, \end{array}$$
where $D\left(t,s,u\right):=\sigma \left(t,s,u\right)\sigma^{⊤}\left(t,s,u\right)$. $w^{\prime }\left(t,s\right)$ is the solution of the following Hamilton–Jacobi–Bellman (HJB) equation driven by $u^{\prime }$:(16)$$\begin{array}{c} -\frac{\partial w^{\prime }\left(t,s\right)}{\partial t}=H\left(t,s,u^{\prime },w^{\prime }\right), \end{array}$$
where $w^{\prime }\left(T,s\right)=g\left(s\right)$. p(t,s) is the solution of the following Fokker–Planck (FP) equation driven by *u*:(17)$$\begin{array}{c} \frac{\partial p(t,s)}{\partial t}=L_{u}^{†}p\left(t,s\right), \end{array}$$
where $p\left(0,s\right)=p_{0}\left(s\right)$. $L_{u}^{†}$ is the forward diffusion operator, which is defined as follows:(18)$$\begin{array}{cc} L_{u}^{†}p\left(t,s\right) & :=-\sum_{i=1}^{d_{s}}\frac{\partial (b_{i}\left(t,s,u\right)p\left(t,s\right))}{\partial s_{i}}+\frac{1}{2}\sum_{i,j=1}^{d_{s}}\frac{\partial^{2}\left(D_{ij}\left(t,s,u\right)p\left(t,s\right)\right)}{\partial s_{i}\partial s_{j}}. \end{array}$$
$L_{u}^{†}$ is the conjugate of $L_{u}$ as follows:(19)$$\begin{array}{c} \int w\left(t,s\right)L_{u}^{†}p\left(t,s\right)ds=\int p\left(t,s\right)L_{u}w\left(t,s\right)ds. \end{array}$$
We derive Equation (Equation 13) in Section C.1.

### 3.2. Necessary Condition

In this subsection, we show the necessary condition of the optimal control function of ML-POSC. It corresponds to Pontryagin’s minimum principle on the probability density function space (Figure 2 (bottom right)). If $u^{*}$ is the optimal control function of ML-POSC (Equation 12), then the following equation is satisfied:(20)$$\begin{array}{c} u^{*}\left(t,z\right)=\arg\min_{u}E_{p_{t}^{*}\left(x|z\right)}\left[H\left(t,s,u,w^{*}\right)\right],\;a.s.{\;}^{\forall }t\in \left[0,T\right],{\;}^{\forall }z\in R^{d_{z}}, \end{array}$$
where $w^{*}\left(t,s\right)$ is the solution of the following HJB equation driven by $u^{*}$:(21)$$\begin{array}{c} -\frac{\partial w^{*}\left(t,s\right)}{\partial t}=H\left(t,s,u^{*},w^{*}\right), \end{array}$$
where $w^{*}\left(T,s\right)=g\left(s\right)$. $p_{t}^{*}\left(x|z\right):=p^{*}\left(t,s\right)/\int p^{*}\left(t,s\right)dx$ is the conditional probability density function of state *x* given memory *z*, and $p^{*}\left(t,s\right)$ is the solution of the following FP equation driven by $u^{*}$:(22)$$\begin{array}{c} \frac{\partial p^{*}\left(t,s\right)}{\partial t}=L_{u^{*}}^{†}p^{*}\left(t,s\right), \end{array}$$
where $p^{*}\left(0,s\right)=p_{0}\left(s\right)$. We derive this result in Section C.2.

In deterministic control, Pontryagin’s minimum principle can be expressed by the derivatives of the Hamiltonian (Figure 2 (bottom left)). Similarly, the system of HJB-FP Equations (Equation 21) and (Equation 22) can be expressed by the variations of the expected Hamiltonian
(23)$$\begin{array}{c} \bar{H}\left(t,p,u,w\right):=E_{p(s)}\left[H\left(t,s,u,w\right)\right] \end{array}$$
as follows: (24)$$\begin{array}{cc} \frac{\partial p^{*}\left(t,s\right)}{\partial t} & =\frac{\delta \bar{H}\left(t,p^{*},u^{*},w^{*}\right)}{\delta w}\left(s\right), \end{array}$$(25)$$\begin{array}{cc} -\frac{\partial w^{*}\left(t,s\right)}{\partial t} & =\frac{\delta \bar{H}\left(t,p^{*},u^{*},w^{*}\right)}{\delta p}\left(s\right), \end{array}$$
where $p^{*}\left(0,s\right)=p_{0}\left(s\right)$ and $w^{*}\left(T,s\right)=g\left(s\right)$ (Figure 2 (bottom right)). Therefore, the system of HJB-FP equations can be interpreted via Pontryagin’s minimum principle on the probability density function space.

### 3.3. Sufficient Condition

Pontryagin’s minimum principle (Equation 20) is only a necessary condition and generally not a sufficient condition. Pontryagin’s minimum principle (Equation 20) becomes a necessary and sufficient condition if the expected Hamiltonian $\bar{H}\left(t,p,u,w\right)$ is convex with respect to *p* and *u*. We obtain this result in Section C.3.

### 3.4. Relationship with Bellman’s Dynamic Programming Principle

From Bellman’s dynamic programming principle on the probability density function space (Figure 2 (top right)) [14], the optimal control function of ML-POSC is given by the following equation:(26)$$\begin{array}{c} u^{*}\left(t,z,p\right)=\arg\min_{u}E_{p(x|z)}\left[H\left(t,s,u,\frac{\delta V^{*}\left(t,p\right)}{\delta p}\left(s\right)\right)\right], \end{array}$$
where $V^{*}\left(t,p\right)$ is the value function on the probability density function space, which is the solution of the following Bellman equation:(27)$$\begin{array}{c} -\frac{\partial V^{*}\left(t,p\right)}{\partial t}=E_{p(s)}\left[H\left(t,s,u^{*},\frac{\delta V^{*}\left(t,p\right)}{\delta p}\left(s\right)\right)\right], \end{array}$$
where $V^{*}\left(T,p\right)=E_{p(s)}\left[g(s)\right]$. More specifically, the optimal control function of ML-POSC is given by $u^{*}\left(t,z\right)=u^{*}\left(t,z,p^{*}\right)$, where $p^{*}$ is the solution of the FP Equation (Equation 22).

Because the Bellman Equation (Equation 27) is a functional differential equation, it cannot be solved even numerically. To resolve this problem, the previous work [14] converted the Bellman Equation (Equation 27) into the HJB Equation (Equation 21) by defining
(28)$$\begin{array}{c} w^{*}\left(t,s\right):=\frac{\delta V^{*}\left(t,p^{*}\right)}{\delta p}\left(s\right), \end{array}$$
where $p^{*}$ is the solution of FP Equation (Equation 22). This approach can be interpreted as the conversion from Bellman’s dynamic programming principle (Figure 2 (top right)) to Pontryagin’s minimum principle (Figure 2 (bottom right)) on the probability density function space.

### 3.5. Relationship with Completely Observable Stochastic Control

In the COSC of the extended state, the control $u_{t}$ is determined based on the extended state $s_{t}$ as $u_{t}=u\left(t,s_{t}\right)$. Therefore, in the COSC of the extended state, Pontryagin’s minimum principle on the probability density function space is given by the following equation:(29)$$\begin{array}{c} u^{*}\left(t,s\right)=\arg\min_{u}H\left(t,s,u,w^{*}\right),\;a.s.{\;}^{\forall }t\in \left[0,T\right],{\;}^{\forall }s\in R^{d_{s}}, \end{array}$$
where $w^{*}\left(t,s\right)$ is the solution of the HJB Equation (Equation 21). Because this proof is almost identical to that of Section 3.2, it is omitted in this paper.

While the optimal control function of ML-POSC (Equation 20) depends on the FP equation and the HJB equation, the optimal control function of COSC (Equation 29) depends only on the HJB equation. From this nice property of COSC, Equation (Equation 29) is not only a necessary condition but also a sufficient condition without assuming the convexity of the expected Hamiltonian. We derive this result in Section C.4.

This result is consistent with the conventional result of COSC [15,16,17,18]. Unlike ML-POSC and MFSC, COSC can be solved by Bellman’s dynamic programming principle on the state space. In COSC, Pontryagin’s minimum principle on the probability density function space is equivalent to Bellman’s dynamic programming principle on the state space. Because Bellman’s dynamic programming principle on the state space is a necessary and sufficient condition, Pontryagin’s minimum principle on the probability density function space may also become a necessary and sufficient condition.

## 4. Forward-Backward Sweep Method

In this section, we propose FBSM for ML-POSC and then prove its convergence by employing the interpretation of the system of HJB-FP equations by Pontryagin’s minimum principle introduced in the previous section.

### 4.1. Forward-Backward Sweep Method

In this subsection, we propose FBSM for ML-POSC, which is summarized in Algorithm 1. FBSM is an algorithm to compute the forward FP equation and the backward HJB equation alternately. More specifically, in the initial step of FBSM, we initialize the control function $u_{0:T-dt}^{0}$ and obtain $p_{0:T}^{0}$ by computing the FP equation forward in time from the initial condition. In the backward step, we obtain $w_{0:T}^{1}$ by computing the HJB equation backward in time from the terminal condition and simultaneously update the control function from $u_{0:T-dt}^{0}$ to $u_{0:T-dt}^{1}$ by minimizing the conditional expected Hamiltonian. In the forward step, we obtain $p_{0:T}^{2}$ by computing the FP equation forward in time from the initial condition and simultaneously update the control function from $u_{0:T-dt}^{1}$ to $u_{0:T-dt}^{2}$ by minimizing the conditional expected Hamiltonian. By iterating the backward and forward steps, the objective function of ML-POSC $J[u_{0:T-dt}^{k}]$ monotonically decreases and finally converges to the local minimum at which the control function of ML-POSC $u_{0:T-dt}^{k}$ satisfies Pontryagin’s minimum principle.

Pontryagin’s minimum principle is only a necessary condition of the optimal control function, not a sufficient condition. Therefore, the control function obtained by FBSM is not necessarily the global optimum except in the case where the expected Hamiltonian is convex. Nevertheless, the control function obtained by FBSM is expected to be superior to most control functions because it is locally optimal.

FBSM has been used in deterministic control [32,34,35,38] and MFSC [39,40,41,42]. However, the convergence of FBSM for these problems is not guaranteed because the backward dynamics depend on the forward dynamics even without the optimal control function (Figure 1c,d). In contrast, the convergence of FBSM is guaranteed in ML-POSC because the backward HJB equation does not depend on the forward FP equation without the optimal control function (Figure 1b). More specifically, in FBSM for ML-POSC, the objective function $J[u_{0:T-dt}^{k}]$ monotonically decreases and finally converges to Pontryagin’s minimum principle. In the following subsections, we prove this nice property of FBSM for ML-POSC.
**Algorithm 1: **Forward-Backward Sweep Method (FBSM)//— Initial step —//k←0$p_{0}^{k}\left(s\right)\leftarrow p_{0}\left(s\right)$**for** t=0 to T−dt **do**   Initialize $u_{t}^{k}\left(z\right)$   $p_{t+dt}^{k}\left(s\right)\leftarrow p_{t}^{k}\left(s\right)+L_{u_{t}^{k}}^{†}p_{t}^{k}\left(s\right)dt$**end for****while** $J[u_{0:T-dt}^{k}]$ do not converge **do**   **if** *k* is even **then**     //— Backward step —//     $w_{T}^{k+1}\left(s\right)\leftarrow g\left(s\right)$     **for** t=T−dt to 0 **do**        $u_{t}^{k+1}\left(z\right)\leftarrow \arg\min_{u}E_{p_{t}^{k}\left(x|z\right)}\left[H(t,s,u,w_{t+dt}^{k+1})\right]$        $w_{t}^{k+1}\left(s\right)\leftarrow w_{t+dt}^{k+1}\left(s\right)+H\left(t,s,u_{t}^{k+1},w_{t+dt}^{k+1}\right)dt$     **end for**   **else**     //— Forward step —//     $p_{0}^{k+1}\left(s\right)\leftarrow p_{0}\left(s\right)$     **for** t=0 to T−dt **do**        $u_{t}^{k+1}\left(z\right)\leftarrow \arg\min_{u}E_{p_{t}^{k+1}\left(x|z\right)}\left[H(t,s,u,w_{t+dt}^{k})\right]$        $p_{t+dt}^{k+1}\left(s\right)\leftarrow p_{t}^{k+1}\left(s\right)+L_{u_{t}^{k+1}}^{†}p_{t}^{k+1}\left(s\right)dt$     **end for**   **end if**   k←k+1**end while****return** $u_{0:T-dt}^{k}$

### 4.2. Preliminary

In this subsection, we show an important result in proving the convergence of FBSM for ML-POSC. We suppose that $u_{0:t-dt,t+dt:T-dt}:=\left\{u_{0},...,u_{t-dt},u_{t+dt},...,u_{T-dt}\right\}$ is given and only $u_{t}$ is optimized as follows:(30)$$\begin{array}{c} u_{t}^{*}:=\arg\min_{u_{t}}J\left[u_{0:T-dt}\right]. \end{array}$$
In ML-POSC, $u_{t}^{*}$ can be calculated as follows:(31)$$\begin{array}{c} u_{t}^{*}\left(z\right)=\arg\min_{u_{t}}E_{p_{t}\left(x|z\right)}\left[H\left(t,s,u_{t},w_{t+dt}\right)\right],\;a.s.{\;}^{\forall }z\in R^{d_{z}}, \end{array}$$
where $w_{t+dt}\left(s\right)$ is the solution of the following time-discretized HJB equation driven by $u_{t+dt:T-dt}$:(32)$$\begin{array}{c} w_{\tau }\left(s\right)=w_{\tau +dt}\left(s\right)+H\left(\tau ,s,u_{\tau },w_{\tau +dt}\right)dt,\;\tau \in \left\{t+dt,...,T-dt\right\}, \end{array}$$
where $w_{T}\left(s\right)=g\left(s\right)$. $p_{t}\left(x|z\right):=p_{t}\left(s\right)/\int p_{t}\left(s\right)dx$ is the conditional probability density function of state *x* given memory *z*, and $p_{t}\left(s\right)$ is the solution of the following time-discretized FP equation driven by $u_{0:t-dt}$:(33)$$\begin{array}{c} p_{\tau +dt}\left(s\right)=p_{\tau }\left(s\right)+L_{u_{\tau }}^{†}p_{\tau }\left(s\right)dt,\;\tau \in \left\{0,...,t-dt\right\}, \end{array}$$
where $p_{0}\left(s\right)$. Equation (Equation 31) is obtained by the similar way to Pontyragin’s minimum principle in Section C.5 and also by the time discretization method in Section C.6.

Importantly, $w_{t+dt}$ does not depend on $u_{t}$ in ML-POSC (Figure 3a) while $\lambda_{t+dt}$ and $w_{t+dt}$ depend on $u_{t}$ in deterministic control (Figure 3b) and MFSC (Figure 3c), respectively. Therefore, $u_{t}^{*}$ can be obtained without modifying $w_{t+dt}$ in ML-POSC, which is essentially different from deterministic control and MFSC. From this nice property, the convergence of FBSM is guaranteed in ML-POSC.

### 4.3. Monotonicity

In FBSM for ML-POSC, the objective function is monotonically non-increasing with respect to the update of the control function at each time step. More specifically,
(34)$$\begin{array}{c} J\left[u_{0:t-dt}^{k},u_{t:T-dt}^{k+1}\right]\leq J\left[u_{0:t}^{k},u_{t+dt:T-dt}^{k+1}\right] \end{array}$$
is satisfied in the backward step, and
(35)$$\begin{array}{c} J\left[u_{0:t-dt}^{k+1},u_{t:T-dt}^{k}\right]\geq J\left[u_{0:t}^{k+1},u_{t+dt:T-dt}^{k}\right] \end{array}$$
is satisfied in the forward step. We prove this result in Section C.7. Furthermore, in FBSM for ML-POSC, the objective function is monotonically non-increasing with respect to the update of the control function at each iteration step as follows:(36)$$\begin{array}{c} J\left[u_{0:T-dt}^{k+1}\right]\leq J\left[u_{0:T-dt}^{k}\right]. \end{array}$$
Equation (Equation 36) is obviously satisfied from Equations (Equation 34) and (Equation 35).

### 4.4. Convergence to Pontryagin’s Minimum Principle

We assume that $J[u_{0:T-dt}]$ has a lower bound. From Equation (Equation 36), FBSM for ML-POSC is guaranteed to converge to the local minimum. Furthermore, we assume that if the candidate of $u_{t}^{k+1}$ includes $u_{t}^{k}$, then set $u_{t}^{k+1}$ at $u_{t}^{k}$. Under these assumptions, FBSM for ML-POSC converges to Pontryagin’s minimum principle (Equation 20). More specifically, if $J\left[u_{0:T-dt}^{k+1}\right]=J\left[u_{0:T-dt}^{k}\right]$ holds, $u_{0:T-dt}^{k+1}$ satisfies Pontryagin’s minimum principle (Equation 20). We prove this result in Section C.8.

Therefore, unlike deterministic control and MFSC, in FBSM for ML-POSC, the objective function $J[u_{0:T-dt}^{k}]$ monotonically decreases and finally converges to the local minimum at which the control function $u_{0:T-dt}^{k}$ satisfies Pontryagin’s minimum principle (Equation 20).

## 5. Linear-Quadratic-Gaussian Problem

In this section, we apply FBSM to the LQG problem of ML-POSC [14]. In the LQG problem of ML-POSC, the system of HJB-FP equations is reduced from partial differential equations to ordinary differential equations.

### 5.1. Problem Formulation

In the LQG problem of ML-POSC, the extended state SDE (Equation 8) is given as follows [14]:(37)$$\begin{array}{c} ds_{t}=\left(A\left(t\right)s_{t}+B\left(t\right)u_{t}\right)dt+\sigma \left(t\right)d\omega_{t}, \end{array}$$
where $s_{0}$ obeys the Gaussian distribution $p_{0}\left(s_{0}\right):=N\left(s_{0}\left|\mu_{0},\Lambda_{0}\right.\right)$ where $\mu_{0}$ is the mean vector and $\Lambda_{0}$ is the precision matrix. The objective function (Equation 11) is given as follows:(38)$$\begin{array}{c} J\left[u\right]:=E_{p(s_{0:T};u)}\left[\int_{0}^{T}\left(s_{t}^{⊤}Q\left(t\right)s_{t}+u_{t}^{⊤}R\left(t\right)u_{t}\right)dt+s_{T}^{⊤}Ps_{T}\right], \end{array}$$
where Q(t)⪰O, R(t)≻O, and P⪰O. The LQG problem of ML-POSC is the problem of finding the optimal control function $u^{*}$ that minimizes the objective function J[u] as follows:(39)$$\begin{array}{c} u^{*}:=\arg\min_{u}J\left[u\right]. \end{array}$$

### 5.2. Pontryagin’s Minimum Principle

In the LQG problem of ML-POSC, Pontryagin’s minimum principle (Equation 20) can be calculated as follows [14]:(40)$$\begin{array}{c} u^{*}\left(t,z\right)=-R^{-1}B^{⊤}\left(\Pi K(\Lambda )(s-\mu )+\Psi \mu \right),\;a.s.{\;}^{\forall }t\in \left[0,T\right],{\;}^{\forall }z\in R^{d_{z}}, \end{array}$$
where K(Λ) is defined as follows:(41)$$\begin{array}{c} K\left(\Lambda \right):=\left(\begin{array}{cc} O & \Lambda_{xx}^{-1}\Lambda_{xz}\\ O & I \end{array}\right), \end{array}$$
where μ(t) and Λ(t) are the mean vector and the precision matrix of the extended state, respectively, which correspond to the solution of the FP Equation (Equation 22). We note that $E_{p_{t}\left(z|x\right)}\left[s\right]=K\left(\Lambda \right)\left(s-\mu \right)+\mu$ is satisfied. μ(t) and Λ(t) are the solutions of the following ordinary differential equations (ODEs): (42)$$\begin{array}{cc} \dot{\mu } & =\left(A-BR^{-1}B^{⊤}\Psi \right)\mu , \end{array}$$(43)$$\begin{array}{cc} \dot{\Lambda } & =-{\left(A-BR^{-1}B^{⊤}\Pi K\left(\Lambda \right)\right)}^{⊤}\Lambda -\Lambda \left(A-BR^{-1}B^{⊤}\Pi K\left(\Lambda \right)\right)-\Lambda \sigma \sigma^{⊤}\Lambda , \end{array}$$
where $\mu \left(0\right)=\mu_{0}$ and $\Lambda \left(0\right)=\Lambda_{0}$. Ψ(t) and Π(t) are the control gain matrices of the deterministic and stochastic extended state, respectively, which correspond to the solution of the HJB Equation (Equation 21). Ψ(t) and Π(t) are the solutions of the following ODEs: (44)$$\begin{array}{cc} -\dot{\Psi } & =Q+A^{⊤}\Psi +\Psi A-\Psi BR^{-1}B^{⊤}\Psi , \end{array}$$(45)$$\begin{array}{cc} -\dot{\Pi } & =Q+A^{⊤}\Pi +\Pi A-\Pi BR^{-1}B^{⊤}\Pi +{\left(I-K\left(\Lambda \right)\right)}^{⊤}\Pi BR^{-1}B^{⊤}\Pi \left(I-K\left(\Lambda \right)\right), \end{array}$$
where Ψ(T)=Π(T)=P. The ODE of Ψ (Equation 44) is the Riccati equation [16,17,18], which also appears in the LQG problem of COSC. In contrast, the ODE of Π (45) is the partially observable Riccati equation [14], which appears only in the LQG problem of ML-POSC. The above result is obtained in [14].

The ODE of Ψ (Equation 44) can be solved backward in time from the terminal condition. Using Ψ, the ODE of μ (Equation 42) can be solved forward in time from the initial condition. In contrast, the ODEs of Π (45) and Λ (43) cannot be solved in a similar way as the ODEs of Ψ (Equation 44) and μ (Equation 42) because they interact with each other, which is a similar problem to the system of HJB-FP equations.

### 5.3. Forward-Backward Sweep Method

In the LQG problem of ML-POSC, FBSM is reduced from Algorithm 1 to Algorithm 2. F(Λ,Π) and G(Λ,Π) are defined by the right-hand sides of the ODEs of Λ (43) and Π (45), respectively, as follows:$$\begin{array}{cc} F(\Lambda ,\Pi ) & :=-{\left(A-BR^{-1}B^{⊤}\Pi K\left(\Lambda \right)\right)}^{⊤}\Lambda -\Lambda \left(A-BR^{-1}B^{⊤}\Pi K\left(\Lambda \right)\right)-\Lambda \sigma \sigma^{⊤}\Lambda ,\\ G(\Lambda ,\Pi ) & :=Q+A^{⊤}\Pi +\Pi A-\Pi BR^{-1}B^{⊤}\Pi +{\left(I-K\left(\Lambda \right)\right)}^{⊤}\Pi BR^{-1}B^{⊤}\Pi \left(I-K\left(\Lambda \right)\right). \end{array}$$
This result is obtained in Section C.9. Importantly, in the LQG problem of ML-POSC, FBSM computes the ODEs of Λ (43) and Π (45) instead of the FP Equation (Equation 22) and the HJB Equation (Equation 21).
**Algorithm 2: **Forward-Backward Sweep Method (FBSM) in the LQG problem//— Initial step —//k←0$\Lambda_{0}^{k}\leftarrow \Lambda_{0}$**for** t=0 to T−dt **do**   Initialize $\Pi_{t+dt}^{k}$   $\Lambda_{t+dt}^{k}\leftarrow \Lambda_{t}^{k}+F\left(\Lambda_{t}^{k},\Pi_{t+dt}^{k}\right)dt$**end for****while** $J[u_{0:T-dt}^{k}]$ do not converge **do**   **if** *k* is even **then**     //— Backward step —//     $\Pi_{T}^{k+1}\leftarrow P$     **for** t=T−dt to 0 **do**        $\Pi_{t}^{k+1}\leftarrow \Pi_{t+dt}^{k+1}+G\left(\Lambda_{t}^{k},\Pi_{t+dt}^{k+1}\right)dt$     **end for**   **else**     //— Forward step —//     $\Lambda_{0}^{k+1}\leftarrow \Lambda_{0}$     **for** t=0 to T−dt **do**        $\Lambda_{t+dt}^{k+1}\leftarrow \Lambda_{t}^{k+1}+F\left(\Lambda_{t}^{k+1},\Pi_{t+dt}^{k}\right)dt$     **end for**   **end if**   k←k+1**end while****return** $u_{0:T-dt}^{k}$

## 6. Numerical Experiments

In this section, we verify the convergence of FBSM in ML-POSC by performing numerical experiments on the LQG and non-LQG problems. The setting of the numerical experiments is the same as the previous work [14].

### 6.1. LQG Problem

In this subsection, we verify the convergence of FBSM for ML-POSC by conducting a numerical experiment on the LQG problem. We consider state $x_{t}\in R$, observation $y_{t}\in R$, and memory $z_{t}\in R$, which evolve by the following SDEs: (46)$$\begin{array}{cc} dx_{t} & =\left(x_{t}+u_{t}\right)dt+d\omega_{t}, \end{array}$$(47)$$\begin{array}{cc} dy_{t} & =x_{t}dt+d\nu_{t}, \end{array}$$(48)$$\begin{array}{cc} dz_{t} & =v_{t}dt+dy_{t}, \end{array}$$
where $x_{0}$ and $z_{0}$ obey the standard Gaussian distributions, $y_{0}$ is an arbitrary real number, $\omega_{t}\in R$ and $\nu_{t}\in R$ are independent standard Wiener processes, and $u_{t}=u\left(t,z_{t}\right)\in R$ and $v_{t}=v\left(t,z_{t}\right)\in R$ are the controls. The objective function to be minimized is given as follows:(49)$$\begin{array}{c} J\left[u,v\right]:=E_{p(x_{0:10},y_{0:10},z_{0:10};u,v)}\left[\int_{0}^{10}\left(x_{t}^{2}+u_{t}^{2}+v_{t}^{2}\right)dt\right]. \end{array}$$
Therefore, the objective of this problem is to minimize the state variance with small state and memory controls.

This problem corresponds to the LQG problem, which is defined by (Equation 37) and (Equation 38). By defining $s_{t}:=\left(x_{t},z_{t}\right)\in R^{2}$, ${\tilde{u}}_{t}:=\left(u_{t},v_{t}\right)\in R^{2}$, and ${\tilde{\omega }}_{t}:=\left(\omega_{t},\nu_{t}\right)\in R^{2}$, the SDEs (46)–(48) can be rewritten as follows:(50)$$\begin{array}{cc} \; & ds_{t}=\left(\left(\begin{array}{cc} 1 & 0\\ 1 & 0 \end{array}\right)s_{t}+{\tilde{u}}_{t}\right)dt+d{\tilde{\omega }}_{t}, \end{array}$$
which corresponds to (Equation 37). Furthermore, the objective function (Equation 49) can be rewritten as follows:(51)$$\begin{array}{c} J\left[\tilde{u}\right]:=E_{p(s_{0:10};\tilde{u})}\left[\int_{0}^{10}\left(s_{t}^{⊤}\left(\begin{array}{cc} 1 & 0\\ 0 & 0 \end{array}\right)s_{t}+{\tilde{u}}_{t}^{⊤}{\tilde{u}}_{t}\right)dt\right], \end{array}$$
which corresponds to (Equation 38).

We apply the FBSM of the LQG problem (Algorithm 2) to this problem. $\Pi^{0}\left(t\right)$ is initialized by $\Pi^{0}\left(t\right)=O$. To solve the ODEs of $\Pi^{k}\left(t\right)$ and $\Lambda^{k}\left(t\right)$, we use the fourth-order Runge–Kutta method. Figure 4 shows the control gain matrix $\Pi^{k}\left(t\right)\in R^{2\times 2}$ and the precision matrix $\Lambda^{k}\left(t\right)\in R^{2\times 2}$ obtained by FBSM. The color of each curve represents the iteration *k*. The darkest curve corresponds to the first iteration k=0, and the brightest curve corresponds to the last iteration k=50. Importantly, $\Pi^{k}\left(t\right)$ and $\Lambda^{k}\left(t\right)$ converge with respect to the iteration *k*.

Figure 5a shows the objective function $J[u^{k}]$ with respect to iteration *k*. The objective function $J[u^{k}]$ monotonically decreases with respect to iteration *k*, which is consistent with Section 4.3. This monotonicity of FBSM is the nice property of ML-POSC that is not guaranteed in deterministic control and MFSC. The objective function $J[u^{k}]$ finally converges, and $u^{k}$ satisfies Pontryagin’s minimum principle from Section 4.4.

Figure 5b–d compare the performance of the control function $u^{k}$ at the first iteration k=0 and the last iteration k=50 by performing a stochastic simulation. At the first iteration k=0, the distributions of state and memory are unstable, and the cumulative cost diverges. In contrast, at the last iteration k=50, the distributions of state and memory are stabilized and the cumulative cost is smaller. This result indicates that FBSM improves the performance in ML-POSC.

Although Figure 5b–d look similar to Figure 2d–f in the previous work [14], they are comparing different things. While Figure 5b–d demonstrate the performance improvement by the FBSM iteration, the previous work [14] compares the performance of the partially observable Riccati Equation (45) with that of the conventional Riccati Equation (Equation 44).

### 6.2. Non-LQG Problem

In this subsection, we verify the convergence of FBSM in ML-POSC by conducting a numerical experiment on the non-LQG problem. We consider state $x_{t}\in R$, observation $y_{t}\in R$, and memory $z_{t}\in R$, which evolve by the following SDEs: (52)$$\begin{array}{cc} dx_{t} & =u_{t}dt+d\omega_{t}, \end{array}$$(53)$$\begin{array}{cc} dy_{t} & =x_{t}dt+d\nu_{t}, \end{array}$$(54)$$\begin{array}{cc} dz_{t} & =dy_{t}, \end{array}$$
where $x_{0}$ and $z_{0}$ obey the Gaussian distributions $p_{0}\left(x_{0}\right)=N\left(x_{0}|0,0.01\right)$ and $p_{0}\left(z_{0}\right)=N\left(z_{0}|0,0.01\right)$, respectively. $y_{0}$ is an arbitrary real number, $\omega_{t}\in R$ and $\nu_{t}\in R$ are independent standard Wiener processes, and $u_{t}=u\left(t,z_{t}\right)\in R$ is the control. For the sake of simplicity, memory control is not considered. The objective function to be minimized is given as follows:(55)$$\begin{array}{c} J\left[u\right]:=E_{p(x_{0:1},y_{0:1},z_{0:1};u)}\left[\int_{0}^{1}\left(Q\left(t,x_{t}\right)+u_{t}^{2}\right)dt+10x_{1}^{2}\right], \end{array}$$
where
(56)$$\begin{array}{c} Q\left(t,x\right):=\left\{\begin{array}{cc} 1000 & (0.3\leq t\leq 0.6,0.1\leq |x|\leq 2.0),\\ 0 & (others). \end{array}\right. \end{array}$$
The cost function is high in 0.3≤t≤0.6 and 0.1≤|x|≤2.0, which represents the obstacles. In addition, the terminal cost function is the lowest at x=0, which represents the desirable goal. Therefore, the system should avoid the obstacles and reach the goal with a small control. Because the cost function is non-quadratic, it is a non-LQG problem.

We apply the FBSM (Algorithm 1) to this problem. $u^{0}\left(t,z\right)$ is initialized by $u^{0}\left(t,z\right)=0$. To solve the HJB equation and the FP equation, we use the finite-difference method. Figure 6 shows $w^{k}\left(t,s\right)$ and $p^{k}\left(t,s\right)$ obtained by FBSM at the first iteration k=0 and at the last iteration k=50. From Section C.6, $w^{k}\left(t,s\right)$ is given as follows:(57)$$\begin{array}{c} w^{k}\left(t,s\right)=E_{p(s_{t+dt:1}|s_{t}=s;u^{k})}\left[\int_{t}^{1}\left(Q\left(\tau ,x_{\tau }\right)+{\left(u_{\tau }^{k}\right)}^{2}\right)d\tau +10x_{1}^{2}\right]. \end{array}$$
Because $u^{0}\left(t,z\right)=0$, $w^{0}\left(t,s\right)$ reflects the cost function corresponding to the obstacles and the goal (Figure 6a–e). In contrast, because $u^{50}\left(t,z\right)\neq 0$, $w^{50}\left(t,s\right)$ becomes more complex (Figure 6f–j). In particular, while $w^{0}\left(t,s\right)$ does not depend on memory *z*, $w^{50}\left(t,s\right)$ depends on memory *z*, which indicates that the control function $u^{50}\left(t,z\right)$ is adjusted by the memory *z*. We note that $w^{0}\left(1,s\right)$ (Figure 6e) and $w^{50}\left(1,s\right)$ (Figure 6j) are the same because they are given by the terminal cost function as $w^{0}\left(1,s\right)=w^{50}\left(1,s\right)=10x^{2}$. Furthermore, while $p^{0}\left(t,s\right)$ is a unimodal distribution (Figure 6k–o), $p^{50}\left(t,s\right)$ is a bimodal distribution (Figure 6p–t), which can avoid the obstacles.

Figure 7a shows the objective function $J[u^{k}]$ with respect to iteration *k*. The objective function $J[u^{k}]$ monotonically decreases with respect to iteration *k*, which is consistent with Section 4.3. This monotonicity of FBSM is the nice property of ML-POSC that is not guaranteed in deterministic control and MFSC. The objective function $J[u^{k}]$ finally converges, and its $u^{k}$ satisfies Pontryagin’s minimum principle from Section 4.4.

Figure 7b,c compare the performance of the control function $u^{k}$ at the first iteration k=0 and the last iteration k=50 by conducting the stochastic simulation. At the first iteration k=0, the obstacles cannot be avoided, which results in a higher objective function. In contrast, at the last iteration k=50, the obstacles can be avoided, which results in a lower objective function. This result indicates that FBSM improves the performance in ML-POSC.

Although Figure 7b,c look similar to Figure 3a,b in the previous work [14], they are comparing different things. While Figure 7b,c demonstrate the performance improvement by the FBSM iteration, the previous work [14] compares the performance of ML-POSC with the local LQG approximation of the conventional POSC.

## 7. Discussion

In this work, we first showed that the system of HJB-FP equations corresponds to Pontryagin’s minimum principle on the probability density function space. Although the relationship between the system of HJB-FP equations and Pontryagin’s minimum principle has been briefly mentioned in MFSC [29,30,31], its details have not yet been investigated. We addressed this problem by deriving the system of HJB-FP equations in a similar way to Pontryagin’s minimum principle. We then proposed FBSM to ML-POSC. Although the convergence of FBSM is generally not guaranteed in deterministic control [32,34,35,38] and MFSC [39,40,41,42], we proved the convergence in ML-POSC by noting the fact that the update of the current control function does not affect the future HJB equation in ML-POSC. Therefore, ML-POSC is a special and nice class where FBSM is guaranteed to converge.

Our derivation of Pontryagin’s minimum principle on the probability density function space is formal, not analytical. Therefore, more mathematically rigorous proofs should be pursued in future work. Nevertheless, because our results are consistent with the conventional results of COSC [15,16,17,18], ML-POSC [14], and MFSC [26,27,28,30,31], they would be reliable except for special cases. Furthermore, our results provide a unified perspective on FBSM in deterministic control [32,34,35,38] and the fixed-point iteration method in MFSC [39,40,41,42], which have been studied independently. It clarifies the different properties of ML-POSC from deterministic control and MFSC, which ensures the convergence of FBSM.

The regularized FBSM has recently been proposed in deterministic control, which is guaranteed to converge even in the general deterministic control [44,45]. Our work gives an intuitive reason why the regularized FBSM is guaranteed to converge. In the regularized FBSM, the Hamiltonian is regularized, which makes the update of the control function smaller. When the regularization is sufficiently strong, the effect of the current control function on the future backward dynamics would be negligible. Therefore, the regularized FBSM of deterministic control would be guaranteed to converge for a similar reason to the FBSM of ML-POSC. However, the convergence of the regularized FBSM is much slower because the stronger regularization makes the update of the control function smaller. The FBSM of ML-POSC does not suffer from such a problem because the future backward dynamics already do not depend on the current control function without regularization.

Our work gives a hint about a modification of the fixed-point iteration method to ensure convergence in MFSC. Although the fixed-point iteration method is the most basic algorithm in MFSC, its convergence is not guaranteed [39,40,41,42]. Our work showed that the fixed-point iteration method is equivalent to the FBSM on the probability density function space. Therefore, the idea of regularized FBSM may also be applied to the fixed-point iteration method. More specifically, the fixed-point iteration method may be guaranteed to converge by regularizing the expected Hamiltonian.

In FBSM, we solve the HJB equation and the FP equation using the finite-difference method. However, because the finite-difference method is prone to the curse of dimensionality, it is difficult to solve high-dimensional ML-POSC. To resolve this problem, two directions can be considered. One direction is the policy iteration method [21,46,47]. Although the policy iteration method is almost the same as FBSM, only the update of the control function is different. While FBSM updates the system of HJB-FP equations and the control function simultaneously, the policy iteration method updates them separately. In the policy iteration method, the system of HJB-FP equations becomes linear, which can be solved by the sampling method [48,49,50]. Because the sampling method is more tractable than the finite-difference method, the policy iteration method may allow high-dimensional ML-POSC to be solved. Furthermore, the policy iteration method has recently been studied in MFSC [51,52,53]. However, its convergence is not guaranteed except for special cases in MFSC. In a similar way to FBSM, the convergence of the policy iteration method may be guaranteed in ML-POSC.

The other direction is machine learning. Neural network-based algorithms have recently been proposed in MFSC, which can solve high-dimensional problems efficiently [54,55]. By extending these algorithms, high-dimensional ML-POSC may be solved efficiently. Furthermore, unlike MFSC, the coupling of the HJB-FP equations is limited only to the optimal control function in ML-POSC. By exploiting this nice property, more efficient algorithms may be devised for ML-POSC.

## Figures and Tables

**Figure 1 entropy-25-00208-f001:**
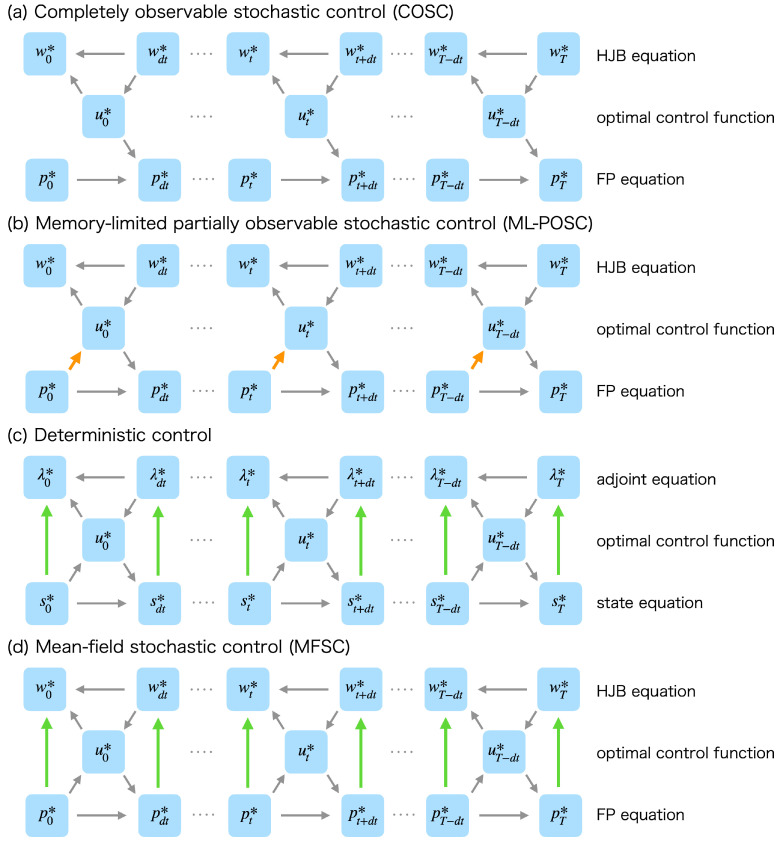
Schematic diagram of the relationship between the backward dynamics, the optimal control function, and the forward dynamics in (**a**) COSC, (**b**) ML-POSC, (**c**) deterministic control, and (**d**) MFSC. $w^{*}$, $p^{*}$, $\lambda^{*}$, and $s^{*}$ are the solutions of the HJB equation, the FP equation, the adjoint equation, and the state equation, respectively. $u^{*}$ is the optimal control function. The arrows indicate the dependence of variables. The variable at the head of an arrow depends on the variable at the tail of the arrow. (**a**) In COSC, because the optimal control function $u^{*}$ depends only on the HJB equation $w^{*}$, it can be obtained by solving the HJB equation $w^{*}$ backward in time from the terminal condition, which is called the value iteration method. (**b**) In ML-POSC, because the optimal control function $u^{*}$ depends on the FP equation $p^{*}$ as well as the HJB equation $w^{*}$ (orange), it cannot be obtained by the value iteration method. In this paper, we propose FBSM for ML-POSC, which computes the HJB equation $w^{*}$ and the FP equation $p^{*}$ alternately. Because the coupling of the HJB equation $w^{*}$ and the FP equation $p^{*}$ is limited only to the optimal control function $u^{*}$, the convergence of FBSM is guaranteed in ML-POSC. (**c**) In deterministic control, because the coupling of the adjoint equation $\lambda^{*}$ and the state equation $s^{*}$ is not limited to the optimal control function $u^{*}$ (green), the convergence of FBSM is not guaranteed. (**d**) In MFSC, because the coupling of the HJB equation $w^{*}$ and the FP equation $p^{*}$ is not limited to the optimal control function $u^{*}$ (green), the convergence of FBSM is not guaranteed.

**Figure 2 entropy-25-00208-f002:**
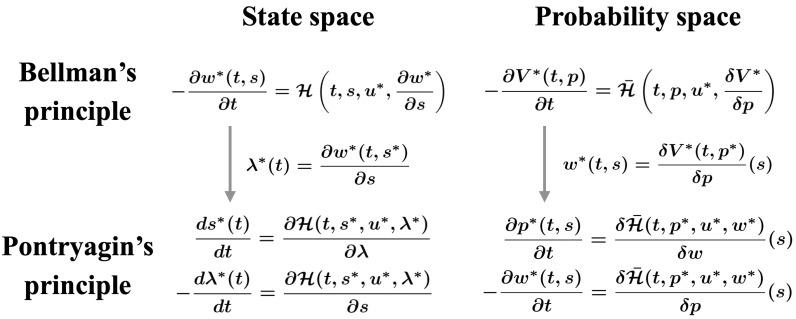
The relationship between Bellman’s dynamic programming principle (**top**) and Pontryagin’s minimum principle (**bottom**) on the state space (**left**) and on the probability density function space (**right**). The left-hand side corresponds to deterministic control, which is briefly reviewed in Appendix A. The right-hand side corresponds to ML-POSC and MFSC, which are shown in Section 3 and Appendix B, respectively. The conventional approach in ML-POSC [14] and MFSC [30,31] can be interpreted as the conversion from Bellman’s dynamic programming principle (**top right**) to Pontryagin’s minimum principle (**bottom right**) on the probability density function space.

**Figure 3 entropy-25-00208-f003:**
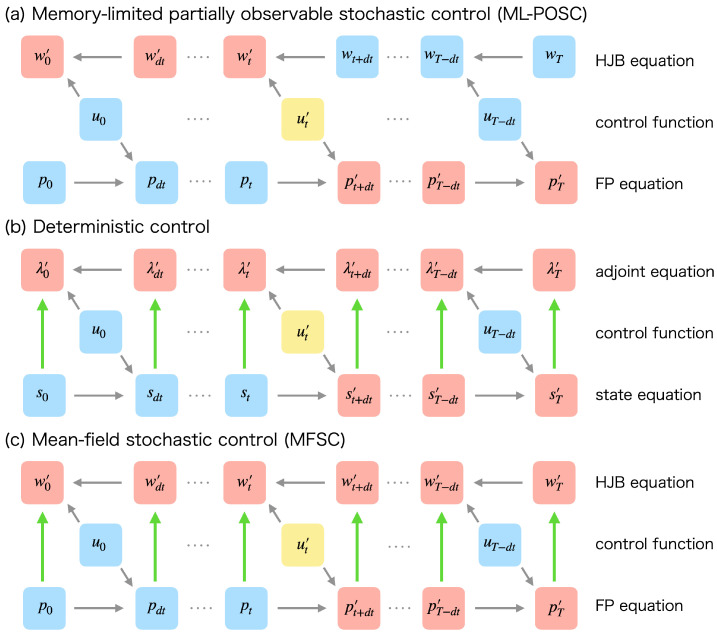
Schematic diagram of the effect of updating the control function to the forward and backward dynamics in (**a**) ML-POSC, (**b**) deterministic control, and (**c**) MFSC. $w_{0:T}$, $p_{0:T}$, $\lambda_{0:T}$, and $s_{0:T}$ are the solutions of the HJB equation, the FP equation, the adjoint equation, and the state equation, respectively. $u_{0:T-dt}$ is a given control function. The arrows indicate the dependence of variables. The variable at the head of an arrow depends on the variable at the tail of the arrow. (**a**) In ML-POSC, while the update from $u_{t}$ to $u_{t}^{\prime }$ (yellow) changes $w_{0:t}$ and $p_{t+dt:T}$ to $w_{0:t}^{\prime }$ and $p_{t+dt:T}^{\prime }$, respectively (red), it does not change $p_{0:t}$ and $w_{t+dt:T}$ (blue). From this property, the convergence of FBSM is guaranteed in ML-POSC. (**b**) In deterministic control, the update from $u_{t}$ to $u_{t}^{\prime }$ (yellow) changes $\lambda_{t+dt:T}$ to $\lambda_{t+dt:T}^{\prime }$ as well (red) because the adjoint equation depends on the state equation (green). Because FBSM does not take into account the change of $\lambda_{t+dt:T}$, the convergence of FBSM is not guaranteed in deterministic control. (**c**) In MFSC, the update from $u_{t}$ to $u_{t}^{\prime }$ (yellow) changes $w_{t+dt:T}$ to $w_{t+dt:T}^{\prime }$ as well (red) because the HJB equation depends on the FP equation (green). Because FBSM does not take into account the change of $w_{t+dt:T}$, the convergence of FBSM is not guaranteed in MFSC.

**Figure 4 entropy-25-00208-f004:**
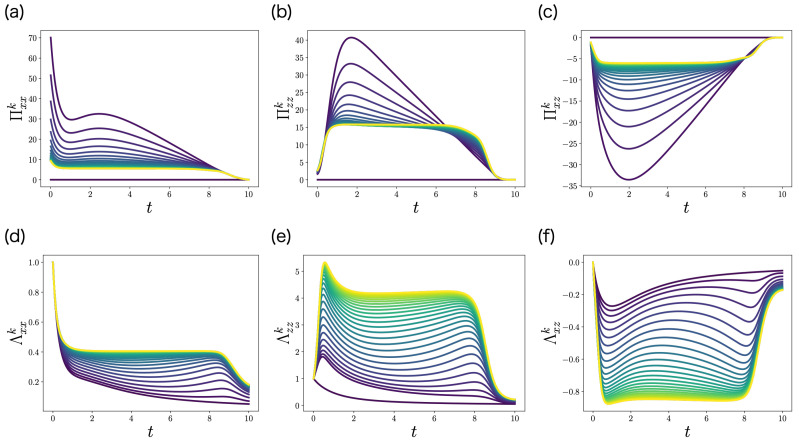
The elements of the control gain matrix $\Pi^{k}\left(t\right)\in R^{2\times 2}$ (**a**–**c**) and the precision matrix $\Lambda^{k}\left(t\right)\in R^{2\times 2}$ (**d**–**f**) obtained by FBSM (Algorithm 2) in the numerical experiment of the LQG problem of ML-POSC. Because $\Pi_{zx}^{k}\left(t\right)=\Pi_{xz}^{k}\left(t\right)$ and $\Lambda_{zx}^{k}\left(t\right)=\Lambda_{xz}^{k}\left(t\right)$, $\Pi_{zx}^{k}\left(t\right)$ and $\Lambda_{zx}^{k}\left(t\right)$ are not visualized. The darkest curve corresponds to the first iteration k=0, and the brightest curve corresponds to the last iteration k=50. $\Pi^{0}\left(t\right)$ is initialized by $\Pi^{0}\left(t\right)=O$.

**Figure 5 entropy-25-00208-f005:**
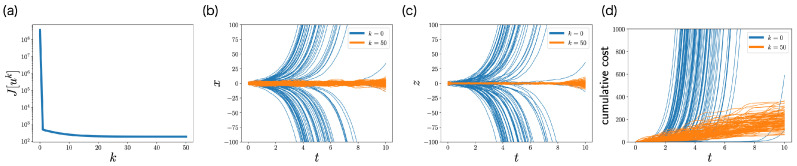
Performance of FBSM in the numerical experiment of the LQG problem of ML-POSC. (**a**) The objective function $J[u^{k}]$ with respect to the iteration *k*. (**b**–**d**) Stochastic simulation of state $x_{t}$ (**b**), memory $z_{t}$ (**c**), and the cumulative cost (**d**) for 100 samples. The expectation of the cumulative cost at t=10 corresponds to the objective function (Equation 49). Blue and orange curves correspond to the first iteration k=0 and the last iteration k=50, respectively.

**Figure 6 entropy-25-00208-f006:**
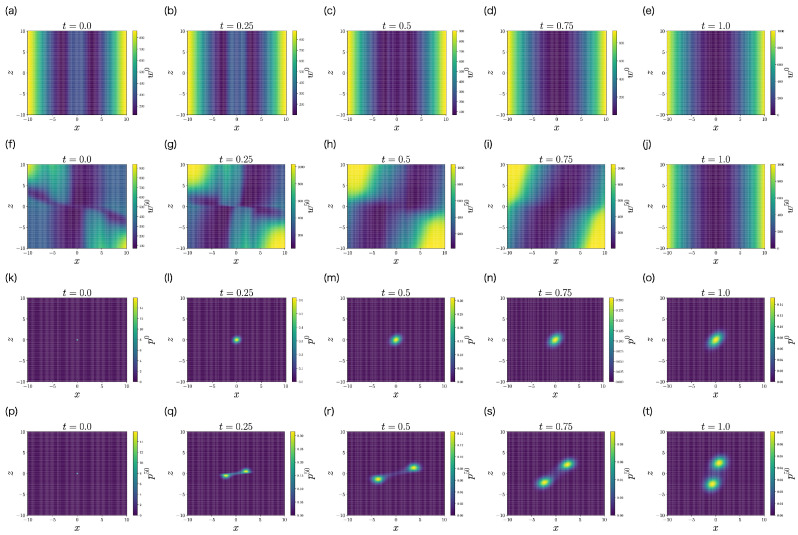
The solutions of the HJB equation $w^{k}\left(t,s\right)$ (**a**–**j**) and the FP equation $p^{k}\left(t,s\right)$ (**k**–**t**) at the first iteration k=0 (**a**–**e**,**k**–**o**) and at the last iteration k=50 (**f**–**j**,**p**–**t**) of FBSM (Algorithm 1) in the numerical experiment of the non-LQG problem of ML-POSC. $u^{0}\left(t,z\right)$ is initialized by $u^{0}\left(t,z\right)=0$.

**Figure 7 entropy-25-00208-f007:**
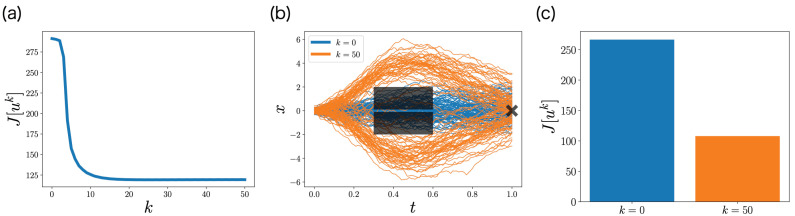
Performance of FBSM in the numerical experiment of the non-LQG problem of ML-POSC. (**a**) The objective function $J[u^{k}]$ with respect to the iteration *k*. (**b**) Stochastic simulation of the state $x_{t}$ for 100 samples. The black rectangles and the cross represent the obstacles and the goal, respectively. Blue and orange curves correspond to the first iteration k=0 and the last iteration k=50, respectively. (**c**) The objective function (Equation 55), which is computed from 100 samples.

## Data Availability

Not applicable.

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
