# Peer review of "Forward-Backward Sweep Method for the System of HJB-FP Equations in Memory-Limited Partially Observable Stochastic Control"

_entropy, 2023, doi:10.3390/e25020208_

Round 1

Reviewer 1 Report

This paper addresses the important problem area of decentralized stochastic control where the agents have limited memory. 

 Unfortunately the paper is not acceptable for publication because  deficiencies in the presentation of the work prevent an evaluation of  its validity. This is despite a large range of scholarly references together with  lengthy derivations involving Dynamic Programming and the Pontryagin Minimum Principle. 

To be specific,  deficiencies in the exposition include:

                      (i) An insufficiently clear problem development. For instance, it is implied that use of systems of the form (3), (10) with  their extended with respect to (1),(2)  permits "memory compression" for limited memory optimization problems (11).   However this is not proven in the paper.

                    (ii)  The form of the Hamiltonian in equations (13)-(14)  is not justified.

                  (iii) One of the central problems with the paper is that expressions such as (24)-(28) contain a probability density as an argument and variations in  the function space of the densities. Since none of this is placed in the required function spaces and none of the function mappings are specified analytically the arguments in the paper are purely formal.

However, the  paper does present interesting and potentially useful computational schemes and these could form the core of a completely revised version of the paper.

Author Response

Dear the reviewer 1,

Thank you very much for your valuable comments and suggestions on our manuscript (entropy-2046432: “Pontryagin's Minimum Principle and Forward-Backward Sweep Method for the System of HJB-FP Equations in Memory-Limited Partially Observable Stochastic Control” by Takehiro Tottori, and Tetsuya J. Kobayashi).

According to your comments, we have thoroughly revised our manuscript. The details of the revisions in our manuscript and our responses to your comments are provided below.

We hope that the current version of our manuscript resolves your comments and concerns.

Yours Sincerely,

Takehiro Tottori & Tetsuya J. Kobayashi,

Details of the revisions in our manuscript and our responses to your comments

  1. An insufficiently clear problem development. For instance, it is implied that use of systems of the form (3), (10) with their extended with respect to (1), (2) permits "memory compression" for limited memory optimization problems (11). However this is not proven in the paper.

The validity and the effectiveness of the formulation of ML-POSC such as (3), (10), and (11) has been shown in our previous work [14]. For example, in the LQG problem of the conventional POSC, it is known that the Kalman filter becomes the optimal compression of the observation history. ML-POSC can reproduce the Kalman filter from the optimal memory dynamics in that problem. This result indicates that ML-POSC is a consistent framework with the conventional POSC. Furthermore, the previous work has demonstrated the effectiveness of ML-POSC in the LQG problem with memory limitation and in the non-LQG problem by numerical experiments. We have added this explanation to Section 2.1 (lines 90-98).

  1. The form of the Hamiltonian in equations (13)(14) is not justified.

The Hamiltonian in equations (13) and (14) is an extension of the conventional Hamiltonian in equation (A4) from the deterministic optimal control problem to the stochastic optimal control problem, which has been used in major textbooks. (For example, reference [16], page 182, equation (3.25). We note that our Hamiltonian is called the generalized Hamiltonian in this reference. Furthermore, their signs are different.) Therefore, the Hamiltonian in equations (13) and (14) is well justified.

  1. One of the central problems with the paper is that expressions such as (24)-(28) contain a probability density as an argument and variations in the function space of the densities. Since none of this is placed in the required function spaces and none of the function mappings are specified analytically the arguments in the paper are purely formal.

As you pointed out, our proofs for Pontryagin’s minimum principle on the probability density function space are purely formal, not analytical. Therefore, more mathematically rigorous proofs remain future challenges. We have added this explanation to our manuscript (For example, lines 130-132, 319-326).

However, we note that our results are consistent with the conventional results with COSC, ML-POSC, and MFSC. Therefore, we believe that our results would be rationalized by a more rigorous treatment of functional analysis with an appropriate setup of function spaces.

Reviewer 2 Report

The content is well written, but the language needs to be revised

Reviewer 3 Report

In the paper "Pontryagin’s Minimum Principle and Forward-Backward Sweep Method for the System of HJB-FP Equations in Memory-Limited Partially Observable Stochastic Control", authors Tottori and Kobayashi elaborate on the relationship between the Hamilton-Jacobi-Bellman Focker-Plank equations and Pontryagin's minimum principle, previously established in mean-filed stochastic control. The authors demonstrate that, unlike in deterministic control problems, the forward-backward sweep method is guaranteed to converge (at least to a local optima) in memory-limited partially observable stochastic control problems, based on the dependence of the backward dynamics on the forward dynamics. This work is closely related to and supports the authors' recent publication in Entropy; "Memory-Limited Partially Observable Stochastic Control and Its Mean-Field Control Approach", indicating strong relevance to this special issue of Entropy. The results presented in this work are interesting and novel.         

Here are some specific comments that I hope may improve the manuscript: 

1) Given the recency of the authors' publication "Memory-Limited Partially Observable Stochastic Control and Its Mean-Field Control Approach" the authors may wish to restate the implications of equation (3) for readers that are not familiar (the "important properties of the memory dynamics"). 

2) I am not sure I understand the combination of directional arrows and colour-coding in Figure 3. Could this be made clearer? 

3) In Figure 6, is subplot (j) correct? It appears very similar to 6(e) and I am somewhat surprised that none of the "structure" of the solution at t=0.75 (Fig. 6i) is visible at t=1.00, particularly given that the corresponding Fokker-Planck  figures 6(o) and 6(t) are different from one-another. If Figure 6(j) is correct, could the authors please elaborate on this. 

4) I note that in Figure 7 in some samples the state at iteration 50 (orange) still passes through the obstacle. Are these trajectories stuck in a local optima, and/or would these trajectories no longer travel through the obstacles if the number of iterations was increased beyond 50. Could the authors please comment on whether this behaviour is a result of the memory limitation, the stochasticity, or something else? 

5) There are a some grammatical oversights that could be corrected, for example: "...superior to most of control functions because..." delete "of", "...an algorithm to ML-POSC..." needs a verb, perhaps "an algorithm to solve ML-POSC problems...", 

Author Response

Dear the reviewer 3,

Thank you very much for your valuable comments and suggestions on our manuscript (entropy-2046432: “Pontryagin's Minimum Principle and Forward-Backward Sweep Method for the System of HJB-FP Equations in Memory-Limited Partially Observable Stochastic Control” by Takehiro Tottori, and Tetsuya J. Kobayashi).

According to your comments, we have thoroughly revised our manuscript. The details of the revisions in our manuscript and our responses to your comments are provided below.

We hope that the current version of our manuscript resolves your comments and concerns.

Yours Sincerely,

Takehiro Tottori & Tetsuya J. Kobayashi,

Details of the revisions in our manuscript and our responses to your comments

  1. Given the recency of the authors' publication "Memory-Limited Partially Observable Stochastic Control and Its Mean-Field Control Approach" the authors may wish to restate the implications of equation (3) for readers that are not familiar (the "important properties of the memory dynamics").

Following your comment, we have added an explanation of the implications of ML-POSC to Section 2.1.

  1. I am not sure I understand the combination of directional arrows and colour-coding in Figure 3. Could this be made clearer?

Following your comment, we have added explanations of the arrows and the colors in Figures 1 and 3 to their captions.

  1. In Figure 6, is subplot (j) correct? It appears very similar to 6(e) and I am somewhat surprised that none of the "structure" of the solution at t=0.75 (Fig. 6i) is visible at t=1.00, particularly given that the corresponding Fokker-Planck figures 6(o) and 6(t) are different from one-another. If Figure 6(j) is correct, could the authors please elaborate on this.

Figure 6(e) and Figure 6(j) are the same because they are given by the terminal cost function. Because the terminal cost function does not depend on the control function, Figure 6(e) and Figure 6(j) are the same even though their control functions are different. We have added this explanation to Section 6.2 (lines 293-294).

  1. I note that in Figure 7 in some samples the state at iteration 50 (orange) still passes through the obstacle. Are these trajectories stuck in a local optima, and/or would these trajectories no longer travel through the obstacles if the number of iterations was increased beyond 50. Could the authors please comment on whether this behaviour is a result of the memory limitation, the stochasticity, or something else?

Increasing the number of iterations beyond 50 does not improve the performance. In addition, FBSM starting from another initial control function converges to the same control function. Therefore, the control function at iteration 50 is probably global optimum although we cannot prove it rigorously.

We think that the system cannot avoid the obstacles with probability 1 in this problem setting probably because it is difficult to perfectly estimate the state of the system from the noisy observation and the limited memory.

  1. There are a some grammatical oversights that could be corrected, for example: "...superior to most of control functions because..." delete "of", "...an algorithm to ML-POSC..." needs a verb, perhaps "an algorithm to solve ML-POSC problems...",

Following your comment, we have revised our grammatical oversights. 

Round 2

Reviewer 1 Report

The problems with this paper which were indicated in this reviewer's first evaluation remain in the revised version.

 The paper is interesting, creative and it is very likely that the computational methods it introduces would be effective. However, the theorems and lemmas are simply not valid theorems and lemmas;  they lack the required rigorous mathematical framework (for instance type information on functions and functions spaces) and the necessary level of mathematical reasoning in the  purported demonstrations of the results.

 I suggest the paper is completely rewritten as a strongly motivated computational methodology with all the so-called theorems presented as a plausible framework for the proposed algorithms.

Reviewer 3 Report

I thank the authors for their efforts in addressing all issues that I raised. 

Author Response

Thank you for your valuable comments on our manuscript. Your comments have improved our manuscript.